# Associations between Nutritional and Immune Status and Clinicopathologic Factors in Patients with Pancreatic Cancer: A Comprehensive Analysis

**DOI:** 10.3390/cancers13205041

**Published:** 2021-10-09

**Authors:** Beata Jabłońska, Krzysztof Pawlicki, Sławomir Mrowiec

**Affiliations:** 1Department of Digestive Tract Surgery, Medical University of Silesia, 40-752 Katowice, Poland; mrowasm@poczta.onet.pl; 2Department of Biophysics, Medical University of Silesia, 40-752 Katowice, Poland; pawlicki@sum.edu.pl

**Keywords:** pancreatic cancer, nutritional status, malnutrition, Nutritional Risk Score, prognostic nutritional index, neutrophil/lymphocyte ratio, monocyte/lymphocyte ratio, platelet/lymphocyte ratio

## Abstract

**Simple Summary:**

This is a comprehensive analysis of the nutritional status (NS) and immune status of 80 pancreatic cancer (PC) patients undergoing curative pancreatic resection. Higher weight loss (WL) was related to the proximal tumor location. Lower serum total protein, albumin, hemoglobin levels, and PNI were reported in older patients. The higher nutritional risk according to NRS 2002 was associated with higher age, higher WL, lower body mass index (BMI), lower total lymphocyte count, longer duration of hospitalization, neoadjuvant chemotherapy, and preoperative biliary drainage. The lower prognostic nutritional index (PNI) was associated with higher WL, lower serum total protein and albumin concentration, lymphocyte count and higher neutrophil/lymphocyte (NLR), monocyte/lymphocyte (MLR), platelet/lymphocyte (PLR) ratios, and duration of hospitalization. In multiple logistic regression analysis, BMI ≥ 30 kg/m^2^ and NRS 2002 ≥ 3 predicted postoperative complications. In multiple linear regression analysis, the higher NRS 2002 score was linked with longer duration of hospitalization and longer duration of postoperative hospitalization was associated with a higher complication rate. Nutritional impairment correlated with a systemic inflammatory response in PC patients. Assessment of nutritional and immune status using basic diagnostic tools and PNI and immune ratio calculation should be the standard management of PC patients before surgery to improve the postoperative outcome.

**Abstract:**

The aim of this study was to assess and analyze the nutritional status (NS) and immune status of pancreatic cancer (PC) patients. The retrospective analysis included 80 PC patients undergoing curative pancreatic resection in the Department of Digestive Tract Surgery of the Medical University (Katowice, Poland). Patients were divided by the tumor location (proximal vs. distal), age (≤65 years vs. >65 years), Nutritional Risk Score 2002 (NRS 2002) (<3 vs. ≥3), prognostic nutritional index (PNI) (<45 vs. ≥45), and the presence of postoperative complications (no-complication vs. complication) as well as the use of neoadjuvant chemotherapy (no neoadjuvant chemotherapy vs. neoadjuvant chemotherapy) into two subgroups, which were compared. Significantly higher weight loss was related to the proximal tumor location (*p* = 0.0104). Significantly lower serum total protein (*p* = 0.0447), albumin (*p* = 0.0468), hemoglobin (*p* = 0.0265) levels, and PNI (*p* = 0.03) were reported in older patients. The higher nutritional risk according to NRS 2002 was significantly associated with higher age (*p* = 0.0187), higher weight loss (*p* < 0.01), lower body mass index (BMI) (*p* = 0.0293), lower total lymphocyte count (*p* = 0.0292), longer duration of hospitalization (*p* = 0.020), neoadjuvant chemotherapy (*p* < 0.01), and preoperative biliary drainage (*p* = 0.0492). The lower PNI was significantly associated with higher weight loss (*p* = 0.0407), lower serum total protein and albumin concentration, lymphocyte count (*p* < 0.01) and higher neutrophil/lymphocyte (NLR), monocyte/lymphocyte (MLR), platelet/lymphocyte (PLR) ratios, and duration of hospitalization (*p* < 0.01). In the multiple logistic regression analysis, BMI ≥ 30 kg/m^2^ (OR: 8.62; 95% CI: 1.24–60.04; *p* = 0.029521) and NRS 2002 ≥ 3 (OR: 2.87; 95% CI: 0.88–9.33; *p* = 0.048818) predicted postoperative complications. In the multiple linear regression analysis, the higher NRS 2002 score was linked with the longer duration of hospitalization (b = 7.67948; *p* = 0.043816), and longer duration of postoperative hospitalization was associated with a higher complication rate (b = 0.273183; *p* = 0.003100). Nutritional impairment correlates with a systemic inflammatory response in PC patients. Obesity (BMI ≥ 30 kg/m^2^) and malnutrition (NRS 2002 ≥ 3) predict postoperative complications, which are associate with a longer hospital stay. Assessment of nutritional and immune status using basic diagnostic tools and PNI and immune ratio (NLR, MLR, PLR) calculation should be the standard management of PC patients before surgery to improve the postoperative outcome.

## 1. Introduction

Pancreatic cancer (PC) is the fourth leading cause of cancer-related mortality in both genders, leading to an all-cause mortality rate of 7% in the world. The 5-year survival rate is 5–8%, with a median survival of 5 months [1,2]. Pancreatic ductal adenocarcinoma (PDAC) is the most common type (95%) [1]. Due to an aggressive tumor biology and late clinical symptoms, the diagnosis of this disease is delayed, which is associated with a poor prognosis [1].

The pancreas plays a crucial role in food digestion and glycemic control. Therefore, PC leads to pancreatic exocrine and endocrine insufficiency. Disturbances of digestion lead to malnutrition reported in up to 80% of PC patients [2]. It is commonly manifested by weight loss (WL), which is secondary to decreased dietary intake due to clinical symptoms including abdominal pain, nausea, anxiety, or depression [2]. Additionally, PC causes duodenal or gastric stenosis by the tumor infiltration or compression leading to the ileus (clinically manifested as nausea and vomiting). In patients with tumors located within the pancreatic head, infiltration or compression on the intrapancreatic common bile duct leads to jaundice and numerous disturbances in bile secretion and bile flow into the duodenum. It is associated with decreased fat digestion and decreased fat-soluble vitamin absorption. Advanced jaundice leads to liver insufficiency. Moreover, in both proximal and distal tumor locations, exocrine pancreatic insufficiency, secondary to the pancreatic duct obstruction, leads to maldigestion and malabsorption of all nutrients. A so-called cancer anorexia–cachexia syndrome reported in PC patients is characterized by anorexia, WL, asthenia, and a poor prognosis [3]. Moreover, PC treatment (surgery, chemotherapy, and radiotherapy) additionally increases the risk of malnutrition in patients. Significant nutritional impairments adversely impact patients’ prognosis, survival, and quality of life (QoL) [4].

On the other hand, the nutritional status (NS) influences the results of PC treatment. It has been reported that impaired NS is associated with higher perioperative morbidity following pancreatectomy and lower survival in PC patients. It has been reported that the poor NS was associated with a higher number of infections, delayed wound healing, impaired blood clotting, and vessel wall fragility and generally a higher number of postoperative complications following pancreatectomy [4,5,6]. It is also known that malnutrition is associated with deteriorated humoral and cellular immune response [6]. Systemic inflammation is considered a significant indicator of a poor prognosis in PC patients [3,6,7,8,9].

According to the literature, malnutrition is reported in 30–50% of hospitalized patients. About 20% of cancer patients die due to malnutrition, not because of the cancer [10,11,12]. Proper assessment of the NS allows the appropriate nutritional therapy in order to support care of PC patients and minimize a risk of postoperative complications. To assess the NS, both objective and subjective criteria are used, including different, anthropometric, clinical, and biochemical parameters [6,10,12].

The aim of the study was to assess and analyze the NS in PC patients using selected anthropometric, clinical and biochemical parameters.

## 2. Materials and Methods

### 2.1. General Study Information: Inclusion and Exclusion Criteria

The retrospective analysis of medical records of 80 PC patients undergoing pancreatectomy in the Department of Digestive Tract Surgery of the Medical University (Katowice, Poland) between January 2018 and March 2021 was performed. Assessment of the NS was performed in patients at the time of hospital admission. There were 40 men and 40 women with a mean age of 65.44 (41–86) years in the analyzed group. Inclusion criteria were as follows: primary PC, age >18 years, and resectable or borderline resectable regionally advanced cancer without confirmed distant metastases prior to surgery. Exclusion criteria were as follows: disseminated cancer (dissemination confirmed before surgery in imaging investigations), cancer recurrence, incomplete demographic, and clinical data. The tumor resectability was determined based on the abdominal and pelvic multidetector computed tomography (CT) performed in the previous 4 weeks before operation [12,13]. 

### 2.2. The Patients’ General Clinical and Pathological Characteristics

#### 2.2.1. Clinical Patients’ Characteristics

The general clinical characteristics of 80 patients are presented in Appendix A. The mean weight recorded before surgery was 70.92 ± 12.92 (40.00–99.00) kg. Mean WL due to the disease was 7.77 ± 8.33 (0.00–30.00). There were 19 (23.75%) patients with WL >10%. In the majority of patients, BMI exceeded 25 kg/m^2^, including 39 (48.75%) patients with BMI 25–30 kg/m^2^ and 5 (6.25%) patients with BMI ≥ 30 kg/m^2^. There were 3 (3.75%) patients with BMI < 18.5 kg/m^2^. The patients’ clinical characteristics regarding medical history, hospitalization, and surgery are presented in Appendix A. There were 58 (72.5%) tumors located within the proximal pancreas and 22 (27.5%) within the distal pancreas. Pancreaticoduodenectomy (PD) was the most frequent surgical procedure performed in 55 (68.75%) patients. Jaundice (52.50%) and abdominal pain (46.25%) were the common clinical symptoms observed in the duration of 4.95 ± 3.90 (1–18) months. The postoperative 30-day morbidity rate was 18.5%, and 30-day mortality rate was 1.25%. Postoperative pancreatic fistula (POPF) (5%) and wound infection (5%) were the most common complications. Reoperations were performed in 9 (11.25%) patients. Twelve (15%) patients underwent pancreatectomy following neoadjuvant chemotherapy, and one (3.25%) had received neoadjuvant radiotherapy before surgery (Appendix A). All patients received perioperative intravenous fluids followed by a standard diet with oral nutritional supplements (ONS). In this study, most of the patients (42.50%) were awarded 2 points in NRS 2002. However, there was also a large number (25.26%) of patients with a score ≥3 in NRS 2002 (Appendix A). The values of laboratory tests are presented in Appendix A. 

#### 2.2.2. Pathological Tumor Characteristics

The pathological tumor analysis is presented in Appendix A. It should be noted that the majority of the analyzed tumors were T2 tumors (72.50%), with metastasis to the lymph nodes (88.75%). There were 9 (11.25%) distal metastasis confirmed in the postoperative histopathological investigation. Adenocarcinoma was the most frequent (93.75%) histological type. Most of the tumors (62.50%) showed a moderate degree of histological differentiation (G2). 

### 2.3. Study Design

All patients were asked about deterioration of the NS (including clinical symptoms such as loss of appetite, jaundice and diarrhea, or constipation, which could potentially affect patient’s food intake), bodyweight before the disease and treatment, unintentional WL, and food intake since the onset of disease. Information on comorbidities (arterial hypertension, ischemic heart disease, type 1 and 2 diabetes mellitus) and smoking (including the amount and duration of smoking and smoking cessation after diagnosis) was collected. The height and weight were measured, and laboratory blood tests were performed at hospital admission. The selected blood counts parameters (hemoglobin and white blood cell (WBC), total lymphocyte, neutrophil, and monocyte counts) and biochemical parameters (serum total protein and albumin, liver and kidney parameters, and cancer serum markers including carcinoembryonic antigen (CEA) and carbohydrate antigen (CA 19.9) were analyzed. The body mass index (BMI) and WL in the course of the disease were calculated. The analyzed hospitalization-related clinical factors included: preoperative biliary drainage, duration of hospitalization, American Society of Anesthesiologists (ASA) classification, duration of the operation, early postoperative complications, and reoperations. The patients were divided into two subgroups according to their BMI into malnourished patients (BMI < 18.5 kg/m^2^) and well-nourished patients (BMI ≥ 18.5 kg/m^2^) as well as four groups according to World Health Organization (WHO) classification [14]. The nutritional risk according to Nutritional Risk Score 2002 (NRS 2002) by the European Society of Parenteral and Enteral Nutrition (ESPEN) was assessed [15,16]. The Onodera’s nutritional prognostic index (PNI) was calculated based on the serum albumin concentration and total lymphocyte count in the peripheral blood by the formula 10 × level of albumin (g/dL) + 0.005 × total lymphocyte count (/mm^3^) [17]. The immunological parameters, such as neutrophil/lymphocyte ratio (NLR), platelet/lymphocyte ratio (PLR), and monocyte lymphocyte ratio (MLR), were calculated [18]. Patients were divided by the tumor location (proximal vs. distal), age (≤ 65 years vs. > 65 years), NRS 2002 (<3 vs. ≥3), PNI (<45 vs. ≥45), and the presence of postoperative complications (no-complication vs. complication), as well as the use of neoadjuvant chemotherapy (no neoadjuvant chemotherapy vs. neoadjuvant chemotherapy) into two subgroups, which were compared. Clinicopathological factors and selected laboratory parameters were compared between the above-mentioned subgroups. Additionally, correlations between selected nutritional parameters (NRS 2002, PNI, BMI) and selected clinicopathological factors were analyzed, and the risk factors for malnutrition and postoperative complications were determined.

The tumors were classified according to the current standard TNM system according to American Joint Commission on Cancer (AJCC) (8th edition) and histological type and grading [19]. Surgical margin status was classified as follows: as the presence of malignant cells (1) directly at the inked surface (R1 direct), (2) within less than 1 mm (R1 ≤ 1 mm), or (3) with a distance greater than 1 mm (R0) [20].

### 2.4. Ethics Approval and Consent to Participate

The Medical University of Silesia Ethics Committee decided that formal consent was unnecessary for this type of study. All procedures performed in studies involving human participants were in accordance with the 1964 Helsinki declaration and its later amendments or comparable ethical standards. 

### 2.5. Statistical Analysis

The Shapiro–Wilk test was used to check for normality of the distribution. The continuous variables were expressed as the means and standard deviations. The categorical variables were presented as numbers and percentages. Depending on the type of statistical distribution, comparisons between groups were performed using the parametric Student’s *t*-test or the non-parametric Mann–Whitney *U* test (for continuous variables) and the *χ^2^* test or the Fisher exact test (for categorical variables). A *p*-value of <0.05 was considered statistically significant. A statistical analysis of correlations between different nutritional parameters (NRS 2002, PNI, BMI) and selected clinicopathologic factors (age, gender, tumor location, histological grading, and clinical stage according to TNM classification), and laboratory parameters was performed using Pearson’s or Spearman’s rank-correlation coefficient, as appropriate. Correlation strength (as a correlation coefficient) and significance (as a *p*-value) were described. A strength coefficient (r) was calculated. The following interpretation of the strength of correlation results was used: 0.00–0.30 (weak correlation), 0.31–0.50 (moderate correlation), 0.51–0.80 (strong correlation), and 0.81–1.00 (very strong correlation). In addition, we evaluated associations between nutritional parameters and clinicopathological factors using a multiple forward stepwise linear regression model analysis. A multiple binomial logistic regression analysis was performed to determine independent factors associated with the prevalence of malnutrition (NRS 2002 ≥ 3) and the presence of postoperative complications. Relative risks were estimated using exposure odds ratios (ORs) and the corresponding 95% confidence intervals (CIs) from cross-tabulation. The statistical analyses were performed using Statistica^®^ software, version 13.3. (StatSoft) (Copyright 1984-2017. TIBCO Software Inc., Statsoft Poland).

## 3. Results

### 3.1. Comparison of Selected Clinicopathological Factors and Nutritional Parameters Depending on the Tumor Location

WL was significantly higher in patients with tumors located within the pancreatic head compared to the distal tumor location (9.37 ± 8.55 kg vs. 2.50 ± 4.88 kg; *p* = 0.0104). Additionally, the percentage of patients with WL >10% was significantly higher in patients with tumors in the proximal location (31.03% vs. 5.55%; *p* = 0326). PNI, NLR, MLR, and PLR were similar in both subgroups. All comparisons between the two general locations are presented in Appendix A.

### 3.2. Comparison of Selected Clinicopathological Factors and Nutritional Parameters Depending on the Age

The analyzed patients were divided into two groups according to the mean age value: ≤65 years (low age group) and >65 years (high age group). In the older patients compared to the younger ones, a significantly higher NRS 2002 score was noted (2.39 ± 1.16 vs. 1.87 ± 1.00; *p* = 0.0326). The older patients had a lower weight (65.86 ± 12.27 kg vs. 76.24 ± 11.47 kg; *p* = 0.002). Serum total protein (5.86 ± 1.05 mg/dL vs. 6.36 ± 0.80 mg/dL; *p* = 0.0447) and albumin (3.33 ± 0.79 mg/dL vs. 3.68 ± 0.62 mg/dL; *p* = 0.0468) concentrations were significantly lower in older patients compared to the younger group. Additionally, PNI was significantly lower in older patients than younger ones (42.96 ± 8.52 vs. 47.77 ± 7.62; *p* = 0.0300). All comparisons between the two age groups are presented in Appendix A.

### 3.3. Comparison of Selected Clinicopathological Factors and Nutritional Parameters Depending on NRS 2002 Classification

Comparison of high and low NRS 2002 groups revealed not significantly more frequent proximal location of the pancreatic tumor in the patients at nutritional risk compared to those with NRS 2002 < 3 (85.71% vs. 67.70%; *p* = 0.0946). There were more older patients in the high-NRS 2002 group compared to the low-NRS 2002 one (71.43% vs. 44.07%; *p* = 0.0312). The weight was significantly lower in patients at nutritional risk compared to those without it (64.74 ± 12.82 kg vs. 73.12 ± 11.31 kg; *p* = 0.0098). Additionally, WL was significantly higher in patients with NRS 2002 ≥ 3 than in patients with NRS 2002 < 3 (13.64 ± 7.69 kg vs. s. 4.93 ± 7.15 kg; *p* = 0.0006). Significantly higher BMI was noted in patients with NRS<3 compared to NRS 2002 ≥ 3 (25.53 ± 3.46 kg/m^2^ vs. 23.66 ± 3.37 kg/m^2^; *p* = 0.0358). Neoadjuvant chemotherapy was more frequent in patients at nutritional risk compared to those with no nutritional risk (33% vs. 8%; *p* = 0.0061). The significant associations between NRS 2002 and age, BMI, and neoadjuvant chemotherapy are partly related to the fact that age, BMI, and disease severity are determinants of NRS 2002 score. Perioperative morbidity rate was comparable in both groups (20% vs. 35%; *p* = 0.1495). The interesting observation was the significantly longer duration of operation in patients at nutritional risk (504.95 ± 120.2 min vs. 430.15 ± 125.47 min; *p* = 0.0201). It was probably associated with the more frequent proximal location requiring PD in the high NRS 2002 group. In laboratory findings, a significantly higher serum level of CEA was noted in patients at nutritional risk (14.79 ± 32.64 ng/mL vs. 3.87 ± 2.86 ng/mL; *p* = 0.0275). There were no statistical differences in most laboratory results between both NRS 2002 groups. The levels of nutritional parameters were not significantly lower and inflammatory parameters were higher in patients at nutritional risk. All laboratory results compared between the two NRS 2002 groups are presented in Appendix A.

### 3.4. Comparison of Selected Clinicopathological Factors and Nutritional Parameters Depending on PNI

The serum total protein (5.28 ± 0.98 mg/dL vs. 6.65 ± 0.47 mg/dL; *p* < 0.0001) and albumin (2.90 ± 0.60 mg/dL vs. 4.01 ± 0.48 mg/dL; *p* < 0.0001) concentrations were significantly lower in the low PNI group compared to the high PNI group. The duration of hospitalization was not significantly longer in patients with PNI <45 compared to those with PNI ≥45 (19.29 ± 13.34 days vs. 14.05 ± 7.22 days; *p* = 0.0741). The morbidity, mortality, and reoperation rates were comparable in both PNI groups. All comparisons between the low and high PNI groups are presented in Appendix A.

### 3.5. Comparison of Selected Clinicopathological Factors and Nutritional Parameters Depending on Presence of Postoperative Complications

The significantly lower total protein concentration (5.75 ± 0.97 mg/dl vs. 6.24 ± 0.94 mg/dl; *p* = 0.0181) and higher C-reactive protein (CRP) concentration (8.33 ± 12.22 mg/L vs. 6.81 ± 10.19 mg/L; *p* = 0.0464) were noted in patients with postoperative complications. The total duration of hospitalization (22.94 ± 15.05 days vs. 12.17 ± 3.56 days; *p* = 0.0001) and hospitalization in the Intensive Care Unit (ICU) (10.25 ± 8.09 days vs. 1.33 ± 1.00 days; *p* = 0.0087) were longer, and the use (38.89% vs. 10.53%; *p* = 0.0056) and duration of postoperative parenteral nutrition (12.11 ± 5.01 days vs. 2.64 ± 4.08 days; *p* = 0.0012) were significantly greater in the complication group compared to patients with no complications. In addition, we compared the duration of postoperative hospitalization according to the cutoff of 10 days (median value) between patients with and without complications (Appendix A). The longer duration of hospitalization (>10 days after surgery) was significantly more frequently reported in the complication group compared to the non-complication group (77.78% vs. 38.71%; *p* = 0.0035, respectively). On the other side, a significantly higher complication rate was noted in patients hospitalized for more than 10 days compared to the shorter hospital stay (36.84% vs. 9.52%; *p* = 0.0035, respectively). Additionally, in Spearman correlation, duration of postoperative hospitalization was positively correlated with postoperative complication rate (r = 0.33, *p* = 0.0031). All comparisons between the two groups are presented in Appendix A.

### 3.6. Comparison of Selected Clinicopathological Factors as Well as Immune and Nutritional Parameters Depending on the Use of Neoadjuvant Chemotherapy

In the divison of our cohort depending on the use of neoadjuvant chemotherapy, we did not find statistical differences in immune parameters and postoperative complications between patients without and with neoadjuvant chemotherapy. It may partly be associated with the fact that only 12/80 patients had received neoadjuvant chemotherapy in our study. There was a statistical difference in NRS 2002 score between two subgroups (2.0294 vs. 2.75 points; *p* = 0.037461), in patients without and with neoadjuvant chemotherapy, respectively, because neoadjuvant chemotherapy is one of the NRS 2002 determinants. The interesting observed phenomenon in our cohort was that all patients who had received neoadjuvant chemotherapy, and only 4.41% of patients without neoadjuvant chemotherapy, had BMI < 18.5 kg/m^2^ (*p* = 0.0000001). The postoperative morbidity, reoperation, and mortality rates were similar in both groups (*p* > 0.05). Although the rate of neoadjuvant chemotherapy in the complication group was almost two folds higher compared to the patients without complications (22.22% vs. 12.90%), the difference was not statistically significant (*p* = 0.3297) (Appendix A). In addition, we compared postoperative complications according to Clavien–Dindo classification between both groups. There was no statistical difference between the two groups (*p* > 0.05). All comparisons between the two groups are presented in Appendix A.

### 3.7. Correlations between Selected Nutritional Parameters and Clinicopathological Factors

A significant positive correlation between NRS 2002 and age (r = 0.2624; *p* = 0.0187), WL (r = 0.6029; *p* < 0.0001), CA 19.9 (r = 0.4776; *p* = 0.045), CEA (r = 0.4573; *p* = 0.049), and duration of hospitalization (r = 0.2623, *p* = 0.020) were noted in our patients. A significant negative correlation was reported between NRS 2002 and BMI (r = −0.2438; *p* = 0.0293).

PNI was positively correlated with serum total protein (r = 0.4779; *p* = 0.0001), and albumin (r = 0.8778; *p* < 0.0001) concentrations as well as total lymphocyte count (r = 0.4726, *p* = 0.0001), and negatively with WL (r = −0.2696, *p* = 0407), duration of hospitalization (r = −0.3082; *p* = 0.0074), as well as NRL (r = −0.3684; *p* = 0.0072), MLR (r = −0.5388; *p* < 0.0001), and PLR (r = −0.4607; *p* = 0.0003, and CA 19.9 (r = −0.1175; *p* = 0.0694).

BMI was negatively correlated with WL (r = −0.2572; *p* = 0.0212), PLT (r = −0.2898; *p* = 0.0096), as well as inflammatory ratios: NLR (r = −0.2795; *p* = 0.0047), and PLR (r = −0.3764; *p* = 0.0036), and MLR (r = −0.2255; *p* = 0.0887).

All correlations are presented in Table 1.

### 3.8. Regression Analysis for Association between Selected Nutritional Parameters and Clinicopathological Factors Including the Impact of NS on Postoperative Outcome

In the multiple linear regression analysis, the higher NRS 2002 score (nutritional risk) was significantly associated with lower serum albumin concentration (b= −0.08421; *p* = 0.024133) and preoperative biliary stenting (b = 0.88227; *p* = 0.0407) (Table 2), while higher serum albumin level was significantly linked with higher PNI (b = 10.3492; *p* = < 0.0001) and higher NLR (b = 1.2694; *p* = 0.020138), as well as lower NRS 2002 classification (b = −1.8304; *p* = 0.003691) and lower MLR (b = −13.7217; *p* = 0.026189) (Table 3).

In the multiple logistic regression analysis, age > 65 years (OR: 7.75, 95% CI: 1.59–37.66; *p* = 0.011127), distal metastases (OR: 20.25, 95% CI: 1.93–212.71; *p* = 0.012159), WL (OR: 5.27, 95% CI: 2.05–13.55; *p* = 0.000565), and neoadjuvant chemotherapy (OR: 12.06, 95% CI: 1.71–84.87; *p* = 0.012399) were significantly associated the higher nutritional risk (NRS 2002 ≥ 3) (Table 4).

In multiple logistic regression analysis, BMI ≥ 30 kg/m^2^ (OR: 8.62; 95% CI: 1.24–60.04; *p* = 0.029521) and NRS 2002 ≥ 3 (OR: 2.87; 95% CI: 0.88–9.33; *p* = 0.048818) predicted postoperative complications (Table 5).

In the multiple linear regression analysis, higher NRS 2002 score was linked with the longer duration of hospitalization (b = 7.67948; *p* = 0.043816) (Table 6), and longer duration of postoperative hospitalization was associated with higher complication rate (b = 0.273183; *p* = 0.003100) (Table 7).

## 4. Discussion

This study showed some relevant associations between the NS and clinicopathological factors. Our comprehensive analysis revealed a higher nutritional risk according to the NRS 2002 in patients with preoperative biliary stenting. In our patients, lower serum albumin concentration as well as higher age, WL, distal metastases, and neoadjuvant chemotherapy were risk factors for malnutrition measured by the NRS 2002. On the other hand, the higher NRS 2002 score and BMI ≥ 30 were significant predictors for postoperative complications in our cohort. Moreover, NRS 2002 ≥ 3 was associated with a longer duration of hospitalization. We noted significant negative correlations between PNI and immune ratios (NLR, MLR, and PLR). Consequently, there is a relationship between various nutritional parameters and clinicopathological factors as well as between NS and the early postoperative outcome.

Numerous publications have shown the association between malnutrition as well as systemic inflammation and a poor prognosis in cancer patients including PC as a result of deterioration of the general condition and a higher number of complications related to the treatment, including major gastrointestinal surgery such as pancreatectomy. Additionally, in PC patients undergoing pancreatectomy, the association between the nutritional/immune status and a prognosis has been reported [2,6,21,22,23,24,25,26,27,28,29,30,31,32,33,34,35]. The negative impact of malnutrition on prognosis in cancer patients is associated with significant immune system dysfunction in both the cellular and humoral immunity [25]. Therefore, malnutrition usually is observed with systemic cancer-related inflammatory response and numerous alterations in immune-inflammatory parameters. Therefore, assessment of the NS is very important because it allows for the selection of malnourished patients with an increased nutritional risk and for taking appropriate nutritional intervention in order to improve patients’ prognosis and reduce complications [10,25].

### 4.1. Obesity as a Risk Factor for PC and Postoperative Morbidity Following Pancreatectomy

In our study, we selected a few simple diagnostic tools to determine patients’ NS. It should be noted that the majority of our patients were found to be overweight or obese (BMI exceeded 25 kg/m^2^), which may confirm the theory that obesity is a risk factor for the development of cancer, including PC [34,36]. Obesity is a very serious and increasing problem worldwide. According to Perrone et al. [36], in 2015, approximately 39% of the world’s adult population was either overweight or obese. Moreover, in 2030, approximately 58% of the world’s population is anticipated to be overweight or obese. At the same time, PC is estimated to become the second leading cause of cancer-related mortality. This confirms the theory that high BMI is strongly associated with an increased PC risk [36]. Moreover, obesity is associated with higher postoperative morbidity and complication rate, which was confirmed in our study, in which the percentage of obese patients was higher in the complication group and BMI≥30 was a risk factor for postoperative morbidity.

### 4.2. Nutritional Risk Factors for Postoperative Morbidity Following Pancreatectomy

Bozzetti et al. [29] reported that pancreatic surgery, advanced age, WL, and low serum albumin were independent risk factors for postoperative complications, while gender, hemoglobin level, and total lymphocytes count were not associated significantly with postoperative complications. In our cohort, a higher NRS 2002 classification, as a marker of malnutrition, and obesity were associated with a higher risk of postoperative complications. The higher NRS 2002 score was also associated with a significantly longer duration of hospitalization in our cohort. Moreover, the incidence of postoperative complications correlated with longer hospital stays in our patients. Moreover, in our study, a significantly higher complication rate has been noted in patients hospitalized for more than 10 days compared to the shorter hospital stay (36.84% vs. 9.52%; *p* = 0.0035, respectively).

### 4.3. Weight and WL in PC Patients

In our study, the mean WL was 7.77 kg and the WL >10% was noted in 23.75% of analyzed patients. According to the literature, up to 80% of patients with pancreatic head cancer present with weight loss at diagnosis [4], with up to 40% having a greater than 10% WL within six months of diagnosis [16]. WL is a recognized cancer marker [4]. In Olsen et al.’s study [37], 21% of patients lost >15% of weight. In Lee et al.’s study [38], which included 499 patients, 40.7% of patients presented involuntary weight loss, and cancer cachexia (weight loss >10%) was manifested in 31.5% of patients. This large study involved a majority (63.5%) of patients with PC located in the pancreatic head that was similar to our patients’ cohort. The better results (lower percentage of patients with cachexia) in our observation may be associated with the fact that our analysis involved only locally or regionally advanced PC. Only 11.25% of our patients were qualified as M1 staging confirmed in the postoperative histopathological finding commonly based on paraaortic lymph node involvement. Moreover, the presence of distal metastases was associated with the higher NRS 2002 classification in our patients. The large study of Lee et al. [38] involved patients in all PC stages as follows: resectable stage (24.2%), locally advanced stage (23.6%), and metastatic stage (52.1%). Additionally, only 23% of patients received surgical resection, and most patients were treated using chemotherapy, radiotherapy, or symptomatic conservative treatment [38]. All patients in our cohort received pancreatectomy.

In another study by Trestini et al. [39] including 73 patients undergoing surgery for PC, the authors noted the mean weight of 73.8 15.1 kg (in our patients 70.92 ± 12.92 kg), with the mean preoperative BMI value of 24.1 ± 4.3 kg/m^2^ (25.04 ± 3.51 kg/m^2^ in our cohort). Only two patients had BMI less than 18.5 kg/m^2^ (three patients in our analysis). This observation regarding only patients undergoing pancreatectomy is comparable with our findings. The other characteristics (mean age, common pancreatectomy type, some laboratory results) were also similar to our observation. WL >10% was reported in 31.5% of patients, which was a higher percentage compared to our cohort. This could be related to a higher number of patients receiving neoadjuvant chemotherapy in Trestini’s study (32.8%) compared to 15% in our cohort. This theory is confirmed by the statistical difference between NRS 2002 in terms of neoadjuvant chemotherapy in our study (33% vs. 8% of our patients received chemotherapy before surgery in the group with or without nutritional risk, respectively). Generally, we noted 26.25% of patients at nutritional risk (NRS 2002 ≥ 3) compared to 80.8% of patients in Trestini’s study. This difference was also associated with the above-mentioned preoperative systemic oncological treatment. The better NS, according to the NRS 2002 score, and lower WL in our patients correlated with lower morbidity (22.5%) compared to 52.1% complication rate in the cited study. In our study and the above-mentioned study, abdominal collections were the most common complications.

### 4.4. Association between Tumor Staging and Nutritional and Immune Parameters in PC Patients

In our study, we did not find the statistical association between the tumor staging and nutritional parameters. This may be associated with the inclusion criteria involving only patients with resectable or borderline resectable PC. According to the literature, metastatic PC is more often correlated with deterioration of nutritional and immune parameters, but not all studies show clear correlation between all nutritional and inflammatory parameters with clinicopathological features. In the study by Lee et al. [38], patients with advanced PC stage more frequently presented LW and cancer cachexia, higher WBC count, and lower lymphocyte count, but serum albumin concentration was not similar in non-metastatic and metastatic disease. Patients with metastatic PC showed significantly lower PNI and higher PLR than those with resectable and locally advanced stage. On the contrary, PLR did not show significant difference according to clinical stages [38]. In the following analysis of three patients’ groups, resectable PC (stage I and II), locally advanced PC (stage III), and metastatic PC (stage IV), the stages I-III were similar according to nutritional and inflammatory parameters, and the results indicated a statistical difference for PC of IV stage. These authors also revealed the statistical association between PNI with numerous laboratory findings such as serum albumin, hemoglobin concentration, lymphocyte (but not WBC), platelet count, NLR, PLR. Our study showed a significant correlation between lower PNI and lower serum total protein, albumin, and higher NLR, MLR, PLR, and CA 19.9. We additionally found a strong association between low PNI and longer duration of hospitalization.

### 4.5. The Prognostic Role of PNI in PC Patients

We divided our patients according to the mean PNI value of 45 and noted significantly lower serum total protein and albumin concentrations and not significantly higher morbidity, mortality, and reoperation rates in the low PNI group. Watanabe et al. [18], in the retrospective analysis of 46 PC patients undergoing PD, demonstrated significantly lower hemoglobin concentrations and significantly higher intraabdominal bleeding in the PNI (<40) group than in the PNI (≥40) group and a significantly higher number of surgical complications greater than grade 3 (according to Clavien–Dindo classification) in the NLR (≥2.5) group.

Another study by Kanda et al. [24], including 268 patients undergoing surgery (including 214 PD) for PC, showed that low preoperative albumin concentration and PNI <45 were significantly associated with postoperative complications. Moreover, low PNI and low BMI were independently associated with the postoperative pancreatic fistula. Generally, III or higher postoperative complications were observed in 32% of patients. In addition, low preoperative PNI (but not low albumin) was an independent prognostic factor for poor survival. These authors also analyzed the impact of preoperative biliary drainage on nutritional status. In this cohort, preoperative biliary drainage was performed in 36.6% of patients due to obstructive jaundice. The median duration of drainage tube placement was 22 (range 2–65) days. The mean preoperative serum albumin concentration and PNI values in both groups were similar. In our cohort, preoperative biliary drainage was performed more frequently (41.25%), and the duration between drainage and operation was longer (mean time of 3.44 months (range 0.5–18)). We noted a statistical association between the use of preoperative biliary drainage and nutritional risk according to NRS 2002 classification. In patients at nutritional risk (NRS 2002 ≥ 3), preoperative biliary stenting was performed significantly more frequently compared to patients with no risk of malnutrition according to NRS 2002 classification (6.19% vs. 52.54%, respectively). This association was also reported in the multiple linear regression analysis. It confirmed the negative impact of preoperative biliary drainage on NS. The duration of preoperative biliary drainage did not impact NRS 2002 in our patients.

### 4.6. Correlations between PNI and Immune Ratios (NLR, MLR, PLR) in PC Patients

The correlations between PNI and immune ratios were analyzed in another retrospective study by Geng et al. [3], which included 321 locally advanced and metastatic PC patients. In this study, PNI correlated negatively with NLR, PLR, and TNF-alfa and positively with LMR (lymphocyte/monocyte ratio). LMR is inversely proportional to MLR, according to our study’s analysis, so we can note a negative correlation between PNI and MLR in Geng’s study. Thus, these correlations are in accordance with our findings. We also noted negative correlations between PNI and all immune ratios (NLR, MLR, and PLR) as well as CA 19.9. These negative correlations confirm the theory regarding deterioration of the NS and systemic inflammation with increased monocyte and platelet count and decreased lymphocyte count in cancer, including PC. Usually, when nutritional parameters decrease, inflammatory markers increase in cancer patients. According to the literature, cancer-associated inflammation, such as increased and defective myelopoiesis, as well as local and systemic inflammation, is strongly associated with tumorigenesis, disease progression, and clinical prognosis [40]. Therefore, our study confirmed Geng et al.’s [3] conclusion regarding a strong association between PNI and the systemic inflammatory response presented by immune ratios (NLR, MLR, and PLR) in PC.

### 4.7. The Optimal Cutoff PNI Value in PC Patients

In the worldwide literature, the cutoff PNI value as a prognostic factor for the postoperative outcome in PC patients ranged from 38 to 48.5 [3,18,24,35,41,42,43,44,45]. Watanabe et al. [18] first established an effective cutoff level in which the difference between both compared groups was greatest. The PNI was tested at set cutoff levels of 40, 45, and 50. Then, the authors divided their group based on the PNI (<40 and ≥40) [18]. In Kanda et al.’s study [24], the PNI value of at least 50 was defined as normal, <50 was regarded as mild malnutrition, <45 as moderate to severe malnutrition, and <40 as serious malnutrition. Finally, the cutoff value of the PNI for clinically significant malnutrition was set at <45 in their study [24]. Onoe et al. [45] used the worst tertile (PNI = 36) as the cutoff value because it specified a subset of patients with poor NS [45]. In Abe et al.’s study [44], the cutoff PNI value was 45, as reported in the literature. In our study, the mean PNI cutoff value of 45 was set, similar to Watanabe et al. [18]. As we mentioned above, lower PNI was significantly associated with higher WL, lower total protein, albumin, lymphocyte, and higher immune rations (NLR, MLR, PLR), and longer duration of hospitalization, which correlates with the literature data [3,18,24,35,41,42,43,44,45].

### 4.8. Association between Age and NS in PC Patients

In our material, the nutritional risk (NRS 2002 ≥ 3) was associated with older age. In a comparison of age between low and high age groups, the significantly higher NRS 2002 classification was noted in patients aged >65 years. The weight was also significantly lower in older patients, and higher age was negatively correlated with weight in our study. This is associated with the fact that age and WL are determinants of the NRS 2002. In addition, the lower serum albumin, total protein, hemoglobin levels, and lower PNI were significantly associated with higher age, which confirms the literature data regarding a poor nutritional status in older patients. Moreover, the significantly more frequent and longer PN administration was related to the poorer nutritional status in older patients compared to younger patients in our cohort. The observation regarding a worse nutritional status in older age is in accordance with the literature. According to the literature, age-related physiological anorexia may predispose patients to protein-energy undernutrition in older patients. Generally, lean body mass decreases with aging, while body fat increases. These factors may decrease energy requirements and intake [46,47].

### 4.9. Association between Tumor Location and NS in PC Patients

To our knowledge, our study was the first to compare nutritional and immune parameters depending on the tumor location within the pancreas. It should be noted that significantly higher WL was reported in patients with PC located within the pancreatic head compared to the distal tumor location. WL >10% was noted in 31.03% vs. 5.55% of patients with tumors located in the proximal and distal pancreas, respectively. It may be associated with a jaundice and biliary stenting in patients with tumors in the pancreatic head. NRS 2002 score was also higher in this patients’ group, but without statistical significance. These findings suggest that the proximal location may be strongly associated with a higher nutritional risk in PC patients.

### 4.10. Association between Neoadjuvant Chemotherapy and NS in PC Patients

As we mentioned above, 33% of patients at nutritional risk vs. 8% in patients without nutritional risk received neoadjuvant chemotherapy, which confirms the fact that preoperative chemotherapy aggravates NS. In our study, we have not found statistical differences in immune parameters and postoperative complications between patients without and with neoadjuvant chemotherapy. It may partly be associated with the fact that only 12/80 patients had received neoadjuvant chemotherapy in our study. There was a statistical difference in NRS 2002 score between two subgroups in patients without and with neoadjuvant chemotherapy because neoadjuvant chemotherapy is one of the NRS 2002 determinants. The interesting phenomenon in our cohort was that all patients who had received neoadjuvant chemotherapy and only 4.41% of patients without neoadjuvant chemotherapy had BMI < 18.5 kg/m^2^. This difference was strongly significant. The problem of neoadjuvant therapy in PC patients undergoing surgery was discussed by Kimura et al. [48]. In this retrospective study, which included 199 patients with borderline resectable PC, the authors showed no association between the high/low preoperative PNI and postoperative complications and significantly better prognosis in patients with high (>42.50) preoperative PNI [44]. So far, the impact of neoadjuvant chemotherapy on the NS and postoperative complications in PC patients has been not clearly recognized, and there is the limited number of relevant studies. Further studies are needed to answer this problem. Despite this fact, in our opinion, it is important to identify patients at nutritional risk following chemotherapy and to introduce nutritional intervention to improve their prognosis.

### 4.11. Strengths and Limitations

In this study, cross-sectional associations between nutritional, clinical, and inflammatory parameters were widely analyzed. There are not a lot of similar studies in the global literature. Our analysis is comprehensive and involves numerous parameters. To our knowledge, this is the first study regarding the association between NS and tumor location in PC patients. This study has some limitations. This is a retrospective, single-center study with a relatively small number of patients. Further prospective, multicenter, large studies are needed to validate our observations.

## 5. Conclusions

Our study confirmed that nutritional impairment correlates with systemic inflammatory response in PC patients. Obesity (BMI ≥ 30 kg/m^2^) and malnutrition (NRS 2002 ≥ 3) predict postoperative complications. On the other hand, higher postoperative morbidity rate is associated with a longer duration of hospitalization. Therefore, assessment of nutritional and immune status using basic diagnostic tools and PNI and immune ratio (NLR, MLR, PLR) calculation should be the standard management of PC patients before surgery to improve the postoperative outcome.

In our opinion, patients with deteriorated nutritional and immune parameters need particular attention during perioperative management. Generally, a longer preoperative nutritional support may be required for PC patients at nutritional risk. Moreover, because of higher WL in patients with the tumor located within the pancreatic head, these patients may require more careful nutritional support in the perioperative period. In addition, due to lower levels of laboratory nutritional parameters, older patients (> 65 years) also need more careful nutritional monitoring and intervention.

## Figures and Tables

**Table 1 cancers-13-05041-t001:** Correlations between NRS 2002, PNI, BMI, and selected clinicopathological and laboratory parameters.

Variable	NRS 2002		PNI		BMI	
	r	*p*	r	*p*	r	*p*
Age	0.2624	0.0187	−0.2077	0.1176	−0.0489	0.6663
Weight loss	0.6029	<0.0001	−0.2696	0.0407	−0.2572	0.0212
BMI	−0.2438	0.0293	0.1798	0.1768	-	-
NRS 2002	-	-	−0.2006	0.1311	−0.2437	0.0293
Duration of hospitalization	0.2603	0.0200	−0.3082	0.0074	−0.0444	0.6953
Total protein	−0.0455	0.6882	0.4779	0.0001	0.0769	0.4979
Albumin	−0.2040	0.0694	0.8778	<0.0001	−0.0884	0.5292
Total lymphocytes	0.1127	0.4000	0.4726	0.0001	0.2464	0.0622
Hemoglobin	−0.1306	0.2480	0.1998	0.1326	0.2743	0.0138
PNI	−0.2006	0.1311	-	-	0.0753	0.6229
NLR	−0.0325	0.8187	−0.3684	0.0072	−0.2795	0.0047
MLR	0.0418	0.7552	−0.5388	<0.0001	−0.2255	0.0887
PLR	0.0448	0.7382	−0.4607	0.0003	−0.3764	0.0036
CA 19.9	0.4776	0.045	−0.4989	0.0694	−0.0031	0.9903
CEA	0.4573	0.049	−0.1175	0.667	−0.2175	0.3710

NRS 2002, Nutritional Risk Score 2002; PNI, prognostic nutritional index; BMI; body mass index; NLR, neutrophil/lymphocyte ratio; MLR, monocyte/lymphocyte ratio; PLR, platelet/lymphocyte ratio; CEA, carcinoembryonic antigen; CA 19.9, carbohydrate antigen; r, Pearson’s/Spearman’s rank-correlation coefficient. Significant correlation (*p* < 0.05) is highlighted in red print.

**Table 2 cancers-13-05041-t002:** Association between NRS 2002 and selected clinicopathological factors and laboratory parameters in multiple regression linear analysis.

**Model 1**	**NRS 2002** **R = 0.61041441 R^2^ = 0.37260575 Adjusted R2= 0.25496932** **F(6.32) = 3.1674 *p* < 0.01489 Standard Error of Estimate: 1.1016**				
**Variable**	**b**	**Standard** **Error** **of b**	***p*-Value**	**r**	***p*-Value**
Intercept	0.52943	2.395166	0.826465		
Albumin	−0.08421	0.035573	0.024133	−0.33147	0.024133
Biliary stenting	0.88227	0.413646	0.0407	0.298654	0.0407
PNI groups <45 vs. >45	0.77489	0.526014	0.150482	0.206272	0.150482
Age	0.03646	0.02173	0.103156	0.234909	0.103156
Lymph node invasion N0 vs. N+	−1.21852	0.905103	0.187672	−0.188508	0.187672
Hemoglobin	0.12867	0.122136	0.299996	0.147515	0.299996
**Model 2**	**NRS 2002** **R = 0.40506490 R^2^ = 0.16407757 Adjusted R2 = 0.13108064** **F(3.76) = 4.9725 *p* < 0.00333 Standard Error of Estimate: 1.0355**				
**Variable**	**b**	**Standard Error of b**	***p*-Value**	**r**	***p*-Value**
Intercept	1.180679	1.231398	0.340694		
Albumin	−0.033543	0.016302	0.043057	−0.215794	0.043057
Biliary stenting	0.561976	0.24273	0.023301	0.242812	0.023301
Age	0.02754	0.014375	0.049158	0.200916	0.049158

NRS 2002, Nutritional Risk Score 2002; PNI, prognostic nutritional index; r, semipartial correlation. Significant association (*p* < 0.05) is highlighted in red print.

**Table 3 cancers-13-05041-t003:** Association between serum albumin concentration and selected clinicopathological factors and laboratory parameters in multiple regression linear analysis.

**Model 1**	**Albumin** **R = 0.83469799; R^2^ = 0.69672073; Adjusted R2 = 9.66104082** **F(6.32) = 3.1674; *p* < 0.01489** **Standard Error of Estimate: 1.1016**				
**Variable**	**b**	**Standard** **Error** **of b**	***p*-Value**	**r**	***p*-Value**
Intercept	18.3744	5.52904	0.002186		
PNI groups <45 vs. >45	10.859	1.522365	<0.0001	0.660446	<0.0001
NRS 2002	−1.5923	0.595842	0.011611	−0.24744	0.011611
Tumor (T)	2.4223	1.57154	0.13276	0.142718	0.13276
NLR	1.2282	0.511082	0.022037	0.222505	0.022037
MLR	−12.6804	5.825692	0.036766	−0.20154	0.036766
**Model 2**	**Albumin** **R = 0.84681110; R^2^ = 0.71708904; Adjusted R2 = 0.67422374** **F(5.33) = 16.729; *p* < 0.00000** **Standard error of estimate: 4.4633**				
**Variable**	**b**	**Standard** **Error** **of b**	***p*-Value**	**r**	***p*-Value**
Intercept	24.8288	3.682835	<0.0001		
PNI groups <45 vs. >45	10.3492	1.515783	<0.0001	0.644842	<0.0001
NRS 2002	−1.8304	0.586996	0.003691	−0.294509	0.003691
NLR	1.2694	0.520608	0.020138	0.23028	0.020138
MLR	−13.7217	5.902307	0.026189	−0.219567	0.026189

NRS 2002, Nutritional Risk Score 2002; PNI, prognostic nutritional index; NLR, neutrophil/lymphocyte ratio; MLR, monocyte/lymphocyte ratio; r, semipartial correlation. Significant association (*p* < 0.05) is highlighted in red print.

**Table 4 cancers-13-05041-t004:** Association between NRS 2002 classification and selected clinicopathological factors and laboratory parameters in multiple binomial regression analysis.

Variable	b	SE b	−95% CI	+95% CI	Wald Stat.	*p*-Value	OR	−95% CI	+95% CI
Intercept	−5.566193	2.566567	−10.596571	−0.535815	4.703395	0.030103	0.003825	0.000025	0.585192
Tumor location	−1.807345	1.090339	−3.94437	0.32968	2.747636	0.097398	0.164089	0.019363	1.390523
Age groups ≤65 vs. >65	2.047763	0.806626	0.466806	3.628721	6.444886	0.011127	7.750546	1.594892	37.664613
Tumor staging (T)	0.493388	0.755004	−0.986392	1.973168	0.42705	0.51344	1.637856	0.37292	7.193428
Lymph node invasion (N)	−1.700343	1.17609	−4.005436	0.604751	2.09022	0.148245	0.182621	0.018216	1.830796
Distal metastases (M)	3.008428	1.199774	0.656913	5.359943	6.28753	0.012159	20.255535	1.928829	212.712803
Smoking	0.667835	0.782185	−0.865219	2.200889	0.728987	0.393212	1.950012	0.420959	9.033043
Weight loss% groups	1.661909	0.482011	0.717185	2.606634	11.887779	0.000565	5.269362	2.048658	13.553349
Neoadjuvant chemotherapy	2.489666	0.995638	0.538252	4.441081	6.252871	0.012399	12.057251	1.71301	84.866596

NRS 2002, Nutritional Risk Score 2002; Binomial logistic regression analysis: SE, standard error; OR, Odds Ratio; 95% CI, 95 percent confidence intervals. Significant association (*p* < 0.05) is highlighted in red print.

**Table 5 cancers-13-05041-t005:** Significant predictors of postoperative complications in multiple binomial logistic regression analysis.

Variable	b	SE b	−95% CI	+95% CI	Wald Stat.	*p*-Value	OR	−95% CI	+95% CI
Intercept	−3.903865	1.191759	−6.23967	−1.56806	10.730317	0.001054	0.020164	0.00195	0.208449
BMI groups <30 vs. ≥30	2.154665	0.989986	0.214328	4.095002	4.736977	0.029521	8.625	1.239029	60.03947
NRS 2002 groups <3 vs. ≥3	1.056053	0.600853	−0.121598	2.233704	3.089115	0.048818	2.875	0.885504	9.334375
Neoadjuvant chemotherapy0 vs. 1	−0.60343	1.214431	−2.98367	1.776811	0.246892	0.619272	0.546933	0.050607	5.910974

BMI; body mass index; NRS 2002, Nutritional Risk Score 2002; Binomial logistic regression analysis: SE, standard error; OR, Ods Ratio; 95% CI, 95 per cent confidence intervals. Significant association (*p* < 0.05) is highlighted in red print.

**Table 6 cancers-13-05041-t006:** Predictors for duration of hospitalization in multiple linear regression analysis.

Model 1	Duration of Postoperative Hospitalization R = 0.40222591; R^2^ = 0.16178568; Adjusted R2 = 0.12089718 F(2.41) = 3.9568; *p* < 0.02684; Standard Error of Estimate: 10.589				
Variable	b	Standard Error of b	*p*-Value	r	*p*-Value
Intercept	22.26649	5.538050	0.000243		
NRS 2002 groups <3 vs. ≥3	7.67948	3.692033	0.043816	0.297407	0.043816
Total protein <6.0 vs. ≥6.0 g/dL	−5.78857	3.251591	0.082451	−0.254543	0.082451

NRS 2002, Nutritional Risk Score 2002; r, semipartial correlation. Significant association (*p* < 0.05) is highlighted in red print.

**Table 7 cancers-13-05041-t007:** Association between complication rate and duration of postoperative hospitalization in multiple regression linear analysis.

Model 1	Complications R = 0.32669159; R^2^ = 0.10672739; Adjusted R2 = 0.09527518; F(1.78) = 9.3194; *p* < 0.00310 Standard Error of Estimate: 0.39970				
Variable	b	Standard Error of b	*p*-Value	r	*p*-Value
Intercept	0.095238	0.107015	0.126588		
Duration of postoperative hospitalization ≤10 vs. >10 days	0.273183	0.089487	0.003100	0.326691	0.003100

Significant association (*p* < 0.05) is highlighted in red print.

## Data Availability

The data presented in this study are available in this article and Appendix A available online.

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
