# Peer review of "Associations between Nutritional and Immune Status and Clinicopathologic Factors in Patients with Pancreatic Cancer: A Comprehensive Analysis"

_cancers, 2021, doi:10.3390/cancers13205041_

Round 1
Reviewer 1 Report
In this study, Jablonska et al. analyzed the nutritional and immune status in pancreatic cancer patients, and reported an association with clinicopathologic parameters, including duration of hospitalization and postoperative complications. The study is meaningful, but several concerns undermine the impact of this study.
First and foremost, previous publications on the prognostic values of nutritional score, including NRS2002, significantly undermined the novelty of this study.
It is unclear at what point was NRS score (or other parameters including PNI) taken. Certain parameters require weekly update; it is unclear whether changes in these parameters would change the authors’ conclusions. Also, are the changes themselves of prognostic values?
Certain correlation analyses were problematic. For example, NRS was scored based partly on BMI, age, and disease severity. Thus correlation between NRS and such parameters was not meaningful.
Writing of the manuscript should be restructured. The results section was intermingled with the methods section, while the method section itself does not stand alone.
Moreover, statistics were thrown without integration. This was manifested in the abstract section that appears to be a hodgepodge assembly of facts.
Author Response
Dear Editors,
Dear Reviewers,
Thank you for peer reviewing of our manuscript cancers-1367776, entitled " Associations between Nutritional and Immune Status and Clinicopathologic Factors in Patients with Pancreatic Cancer: a Comprehensive Analysis".
Thank you for your questions and comments. We have fully addressed all the comments and my responses appear below. Our revised work includes corrections according to reviewers’ comments in the text. Our revisions, made according to reviewers’ comments, are highlighted in the green print.
We take this opportunity to express my gratitude to the reviewers for their constructive and useful remarks. Their comments allowed us to identify areas in my manuscript that needed modification.
We also thank you for allowing me to resubmit a revised copy of the manuscript.
We hope that the revised manuscript is now acceptable for publication in Cancers.
Yours sincerely,
Beata Jabłońska.
Responses to Reviewers’ comments
Reviewer 1
In this study, Jablonska et al. analyzed the nutritional and immune status in pancreatic cancer patients, and reported an association with clinicopathologic parameters, including duration of hospitalization and postoperative complications. The study is meaningful, but several concerns undermine the impact of this study.
Comment:
First and foremost, previous publications on the prognostic values of nutritional score, including NRS2002, significantly undermined the novelty of this study.
Answer:
Thank you for your comment. We agree that the prognostic values of nutritional score, including NRS 2002, have been assessed in previous publications, but there are not numerous publications including such comprehensive analysis of various nutritional and immune parameters and clinicopathological factors in PC patients. In our study, we performed a comprehensive analysis including combined nutritional and immune parameters. We analyzed not only prognostic values of nutritional parameters but also complex associations between them and clinicopathological patient- and tumor related factors. In addition, we compared these parameters between two location of pancreatic cancer (proximal and distal). We have not found such precise comparison regarding tumor location in previous publications. To our knowledge, this is the first such study on this subject.
Moreover, our study is a Polish / Central European voice in the worldwide discussion regarding this subject. It may be used in a further meta-analysis. So far, there are only singular meta-analyses regarding the prognostic role of the hematological and immune parameters in PC patients. So far, complex associations between nutritional parameters and clinicopathological factors have been not clearly determined.
In conclusion, our study included a detailed and comprehensive comparative analysis of numerous nutritional and immune parameters. Taking into account all the above-mentioned arguments, the novelty of our study is considerable.
Comment:
It is unclear at what point was NRS score (or other parameters including PNI) taken. Certain parameters require weekly update; it is unclear whether changes in these parameters would change the authors’ conclusions. Also, are the changes themselves of prognostic values?
Answer:
Thank you for your comment. All analyzed nutritional and immune parameters were taken at hospital admission. This information was present in the methods section, paragraph 2.3. as follows:
The height and weight were measured, and laboratory blood tests were performed at hospital admission.
The aim of our study was to determine associations between preoperative nutritional and immune parameters and clinicopathological factors as well as to assess the impact of preoperative nutritional and immune parameters on postoperative outcome. The postoperative changes of parameters were not subjects of our study.
Comment:
Certain correlation analyses were problematic. For example, NRS was scored based partly on BMI, age, and disease severity. Thus correlation between NRS and such parameters was not meaningful.
Answer:
Thank you for this comment. We agree that BMI, age, and disease severity are determinants of the NRS 2002, but the summarized pointing in NRS 2002 score depends equally on the patient’s nutritional status and food intake problems and the impact of disease severity on nutritional status. Therefore, this association is not simple. Although age over 70 years is considered as a risk factor, and is included in the screening tool as well, giving 1 point, in younger patients (age<70 years), different age may also impact on nutritional status despite no points for age in the NRS 2002. Thus, we decided to include this association in the text. Taking into account this limitation, we have presented the explanation that age and weight loss are determinants of the NRS 2002 score in the results section (paragraph 3.3.) as follows:
The significant associations between NRS 2002 and age, BMI, and neoadjuvant chemotherapy are partly related to the fact that age, BMI, and disease severity are determinants of NRS 2002 score.
Comment:
Writing of the manuscript should be restructured. The results section was intermingled with the methods section, while the method section itself does not stand alone.
Answer:
Thank you for your suggestion. The manuscript has been restructured according to your suggestions. The information regarding patients’ characteristics has been included in the methods section and only analyses of nutritional and immune parameters have been included in the results section as follows (pages 2-4):
- Materials and Methods
2.1. General study information: inclusion and exclusion criteria
The retrospective analysis of medical records of 80 PC patients undergoing pancreatectomy in the Department of Digestive Tract Surgery of the Medical University (Katowice, Poland) between January 2018 and March 2021 was performed. Assessment of the NS was performed in patients at the time of hospital admission. There were 40 men and 40 women with mean age of 65.44 (41-86) years in the analyzed group. Inclusion criteria were as follows: primary PC, age > 18 years, resectable or borderline resectable regionally advanced cancer without confirmed distant metastases prior surgery. Exclusion criteria were as follows: disseminated cancer (dissemination confirmed before surgery in imaging investigations), cancer recurrence, incomplete demographic and clinical data. The tumor resectability was determined based on the abdominal and pelvic multidetector computed tomography (CT) performed in the previous 4 weeks before operation [12,13].
2.2. The patients’ general clinical and pathological characteristics
2.2.1. Clinical patients’ characteristics
The general clinical characteristics of 80 patients are presented in Table S1. The mean weight recorded before surgery was 70.92±12.92 (40.00-99.00) kg. Mean WL due to the disease was 7.77±8.33 (0.00-30.00). There were 19 (23.75%) patients with WL >10%. In the majority of patients, BMI exceeded 25 kg/m2 including 39 (48.75%) patients with BMI 25-30 kg/m2 and 5 (6.25%) patients with BMI ≥30 kg/m2. There were 3 (3.75%) patients with BMI <18.5 kg/m2. The patients’ clinical characteristics regarding medical history, hospitalization and surgery was presented in Table S2. There were 58 (72.5%) tumors located within the proximal pancreas and 22 (27.5%) within the distal pancreas. Pancreaticoduodenectomy (PD) was the most frequent surgical procedure performed in 55 (68.75%) patients. Jaundice (52.50%) and abdominal pain (46.25%) were the common clinical symptoms observed in the duration of 4.95±3.90 (1-18) months. The postoperative 30-day morbidity rate was 18.5%, 30-day mortality rate was 1.25%. Postoperative pancreatic fistula (POPF) (5%) and wound infection (5%) were the most common complications. Reoperations were performed in 9 (11.25%) patients. Twelve (15%) patients underwent pancreatectomy following neoadjuvant chemotherapy, and one (3.25%) had received neoadjuvant radiotherapy before surgery (Table S2). All patients received perioperative intravenous fluids followed by a standard diet with oral nutritional supplements (ONS). In this study, most of patients (42.50%) were awarded 2 points in NRS 2002. However, there was also a large number (25.26%) patients with a score ≥ 3 in NRS 2002 (Table S1). The values of laboratory tests were presented in Table S3.
2.2.2. Pathological tumor characteristics
The pathological tumor analysis is presented in Table S4. It should be noted that the majority of the analyzed tumors were T2 tumors (72.50%), with metastasis to the lymph nodes (88.75%). There were 9 (11.25%) of distal metastasis confirmed in the postoperative histopathological investigation. Adenocarcinoma was the most frequent (93.75%) histological type. Most of the tumors (62.50%) showed a moderate degree of histological differentiation (G2).
2.3. Study design
All patients were asked about deterioration of the NS (including clinical symptoms such as loss of appetite, jaundice and diarrhea or constipation which could potentially affect patient’s food intake), body weight before the disease and treatment, unintentional WL, and food intake since the onset of disease. Information on comorbidities (arterial hypertension, ischemic heart disease, type 1 and 2 diabetes mellitus) and smoking (including the amount and duration of smoking and smoking cessation after diagnosis) was collected. The height and weight were measured, and laboratory blood tests were performed at hospital admission. The selected blood counts parameters (hemoglobin and white blood cell (WBC), total lymphocyte, neutrophil, monocyte counts) and biochemical parameters (serum total protein and albumin, liver and kidney parameters and cancer serum markers including carcinoembryonic antigen (CEA) and carbohydrate antigen (CA 19.9) were analyzed. The body mass index (BMI) and WL in the course of the disease were calculated. The analyzed hospitalization-related clinical factors included: preoperative biliary drainage, duration of hospitalization, American Society of Anesthesiologists (ASA) classification, duration of the operation, early postoperative complications, and reoperations. The patients were divided into two subgroups according to their BMI into malnourished patients (BMI <18.5 kg/m2) and well-nourished patients (BMI ≥18.5 kg/m2) as well as four groups according to World Health Organization (WHO) classification [14]. The nutritional risk according to Nutritional Risk Score 2002 (NRS 2002) by the European Society of Parenteral and Enteral Nutrition (ESPEN) was assessed [15,16]. The Onodera’s nutritional prognostic index (PNI) was calculated based on the serum albumin concentration and total lymphocyte count in the peripheral blood by the formula: 10 × level of albumin (g/dl) + 0.005 × total lymphocyte count (/mm3) [17]. The immunological parameters, such as neutrophil/lymphocyte ratio (NLR), platelet/lymphocyte ratio (PLR), and monocyte /lymphocyte ratio (MLR), were calculated [18]. Patients were divided by the tumor location (proximal vs. distal), age (≤ 65 years vs. > 65 years), NRS 2002 (<3 vs. ≥3), PNI (<45 vs. ≥45), and the presence of postoperative complications (no-complication vs. complication) as well as the use of neoadjuvant chemotherapy (no neoadjuvant chemotherapy vs. neoadjuvant chemotherapy) into two subgroups which were compared. Clinicopathological factors and selected laboratory parameters were compared between the above mentioned subgroups. Also, correlations between selected nutritional parameters (NRS 2002, PNI, BMI) and selected clinicopathological factors were analyzed, as well as the risk factors for malnutrition and postoperative complications were determined.
The tumors were classified according to current standard TNM system according to American Joint Commission on Cancer (AJCC) (8th edition) and histological type and grading [19]. Surgical margin status was classified as follows: as the presence of malignant cells (1) directly at the inked surface (R1 direct), (2) within less than 1 mm (R1 ≤ 1 mm), or (3) with a distance greater than 1 mm (R0) [20].
2.4. Ethics approval and consent to participate
The Medical University of Silesia Ethics Committee decided that for this type of study formal consent was unnecessary. All procedures performed in studies involving human participants were in accordance with the 1964 Helsinki declaration and its later amendments or comparable ethical standards.
2.5. Statistical analysis
The Shapiro-Wilk test was used to check for normality of the distribution. The continuous variables were expressed as the means and standard deviations. The categorical variables were presented as numbers and percentages. Depending on the type of statistical distribution, comparisons between groups were performed using the parametric Student’s t-test or the non-parametric Mann-Whitney U test (for continuous variables), and the χ2 test or the Fisher exact test (for categorical variables). A p-value of < 0.05 was considered statistically significant. A statistical analysis of correlations between different nutritional parameters (NRS 2002, PNI, BMI) and selected clinicopathologic factors (age, gender, tumor location, histological grading and clinical stage according to TNM classification) and laboratory parameters was performed using Pearson’s or Spearman’s rank-correlation coefficient, as appropriate. Correlation strength (as a correlation coefficient) and significance (as a p value) were described. A strength coefficient (r) was calculated. The following interpretation of the strength of correlation results was used: 0.00-0.30 (weak correlation), 0.31-0.50 (moderate correlation), 0.51-0.80 (strong correlation), and 0.81-1.00 (very strong correlation). In addition, we evaluated associations between nutritional parameters and clinicopathological factors using a multiple forward stepwise linear regression model analysis. A multiple binomial logistic regression analysis was performed to determine independent factors associated with the prevalence of malnutrition (NRS 2002 ≥3), and the presence of postoperative complications. Relative risks were estimated using exposure odds ratios (ORs) and the corresponding 95% confidence intervals (CIs) from cross tabulation. The statistical analyses were performed using Statistica® software, version 13.3. (StatSoft).
- Results
3.1. Comparison of selected clinicopathological factors and nutritional parameters depending on the tumor location
Comment:
Moreover, statistics were thrown without integration. This was manifested in the abstract section that appears to be a hodgepodge assembly of facts.
Answer:
The statistical analysis has been performed in several steps: descriptive statistics including general patients’ characteristics (1), comparisons of analyzed subgroups using Student, Mann-Whitney or chi-square tests (2), correlation between selected nutritional parameters and clinicopahological and laboratory parameters using basic correlation statististics (3), and advanced statistics using regression analyses (4). This schema, based on the statistic methods, has been used for results presentation to make them more clear. Moreover, in the first step: associations between nutritional / immune parameters and clinicopathological factors have been analyzed and presented, and in the second step: prognostic role of nutritional parameters including postoperative complications and duration of hospitalization. According to your suggestion, we have little changed the order of presented results in the abstract by integration them according to the analyzed parameter (in the following order: tumor location, age, NRS 2002, PNI, postoperative complications, duration of hospitalization) as follows:
Significantly higher weight loss was related to the proximal tumor location (p=0.0104). Significantly lower serum total protein (p=0.0447), albumin (p=0.0468), hemoglobin (p=0.0265) levels and PNI (p=0.03) were reported in older patients. The higher nutritional risk according to NRS 2002 was significantly associated with higher age (p=0.0187), higher weight loss (p<0.01), lower body mass index (p=0.0293), lower total lymphocyte count (p=0.0292), longer duration of hospitalization (p=0.020), neoadjuvant chemotherapy (p<0.01), preoperative biliary drainage (p=0.0492). The lower PNI was significantly associated with higher weight loss (p=0.0407), lower serum total protein and albumin concentration, lymphocyte count (p<0.01) and higher neutrophil/lymphocyte (NLR), monocyte/lymphocyte (MLR), platelet/ /lymphocyte (PLR) ratios, as well as duration of hospitalization (p<0.01). In the multiple logistic regression analysis, body mass index (BMI) ≥30 kg/m2 (OR: 8.62; 95%CI: 1.24-60.04; p=0.029521) and NRS 2002 ≥3 (OR: 2.87; 95%CI: 0.88-9.33; p=0.048818) predicted postoperative complications. In the multiple linear regression analysis, the higher NRS 2002 score was linked with the longer duration of hospitalization (b = 7.67948; p=0.043816), and longer duration of postoperative hospitalization was associated with higher complication rate (b = 0.273183; p=0.003100).

Reviewer 2 Report
This is the article for special issue: Prognostic and Predictive Markers in Pancreatic Cancer. This paper and its content might be suitable for this special issue. However, the analyzed patient’s number was only 80 resected pancreatic cancer patients, which was too small. Also, taking previously published many studies concerning the nutritional and immune status in pancreatic cancer into consideration, this paper lacks the novelty. The comprehensive analysis was conducted at this time, but focus or conclusion of this study is extremely unclear.
Author Response
Dear Editors,
Dear Reviewers,
Thank you for peer reviewing of our manuscript cancers-1367776, entitled " Associations between Nutritional and Immune Status and Clinicopathologic Factors in Patients with Pancreatic Cancer: a Comprehensive Analysis".
Thank you for your questions and comments. We have fully addressed all the comments and my responses appear below. Our revised work includes corrections according to reviewers’ comments in the text. Our revisions, made according to reviewers’ comments, are highlighted in the green print.
We take this opportunity to express my gratitude to the reviewers for their constructive and useful remarks. Their comments allowed us to identify areas in my manuscript that needed modification.
We also thank you for allowing me to resubmit a revised copy of the manuscript.
We hope that the revised manuscript is now acceptable for publication in Cancers.
Yours sincerely,
Beata Jabłońska.
Responses to Reviewers’ comments
Reviewer 2
Comment:
This is the article for special issue: Prognostic and Predictive Markers in Pancreatic Cancer. This paper and its content might be suitable for this special issue. However, the analyzed patient’s number was only 80 resected pancreatic cancer patients, which was too small. Also, taking previously published many studies concerning the nutritional and immune status in pancreatic cancer into consideration, this paper lacks the novelty. The comprehensive analysis was conducted at this time, but focus or conclusion of this study is extremely unclear.
Answer:
Thank you for your comment. The number of analyzed patients is not very large, but this number is sufficient for a relevant statistical analysis. Although our study includes a smaller cohort, it is a Polish / Central European voice in the discussion on this subject. To our knowledge, there are also another valuable studies including similar or smaller patients’ number published in the prestigious journals, for instance:
Regarding articles on the similar subject published in various prestigious journals:
Kimura N, Yamada S, Takami H, Murotani K, Yoshioka I, Shibuya K, Sonohara F, Hoshino Y, Hirano K, Watanabe T, Baba H, Mori K, Miwa T, Kanda M, Hayashi M, Matsui K, Okumura T, Kodera Y, Fujii T. Optimal Preoperative Multidisciplinary Treatment in Borderline Resectable Pancreatic Cancer. Cancers (Basel). 2020;13(1):36. Published 2020 Dec 24. doi:10.3390/cancers13010036: 88 patients
Asaoka T, Miyamoto A, Maeda S, Tsujie M, Hama N, Yamamoto K, Miyake M, Haraguchi N, Nishikawa K, Hirao M, Ikeda M, Sekimoto M, Nakamori S. Prognostic impact of preoperative NLR and CA19-9 in pancreatic cancer. Pancreatology. 2016 May-Jun;16(3):434-40. doi: 10.1016/j.pan.2015.10.006. Epub 2015 Nov 10. PMID: 26852169:46 patients
Trestini I, Paiella S, Sandini M, Sperduti I, Elio G, Pollini T, Melisi D, Auriemma A, Soldà C, Bonaiuto C, Tregnago D, Avancini A, Secchettin E, Bonamini D, Lanza M, Pilotto S, Malleo G, Salvia R, Bovo C, Gianotti L, Bassi C, Milella M. Prognostic Impact of Preoperative Nutritional Risk in Patients Who Undergo Surgery for Pancreatic Adenocarcinoma. Ann Surg Oncol. 2020 Dec;27(13):5325-5334. doi: 10.1245/s10434-020-08515-5. Epub 2020 May 9. PMID: 32388740: 73 patients
Ferrucci LM, Bell D, Thornton J, Black G, McCorkle R, Heimburger DC, Saif MW. Nutritional status of patients with locally advanced pancreatic cancer: a pilot study. Support Care Cancer. 2011 Nov;19(11):1729-34. doi: 10.1007/s00520-010-1011-x. Epub 2010 Oct 22. PMID: 20967470; PMCID: PMC3138878: 14 patients.
Regarding retrospective clinical studies recently publisehd in Cancers journal:
Tomishima K, Ishii S, Fujisawa T, Ikemura M, Ota H, Kabemura D, Ushio M, Fukuma T, Takahashi S, Yamagata W, Takasaki Y, Suzuki A, Ito K, Saito H, Nagahara A, Isayama H. Duration of Reduced CA19-9 Levels Is a Better Prognostic Factor Than Its Rate of Reduction for Unresectable Locally Advanced Pancreatic Cancer. Cancers. 2021; 13(16):4224. https://doi.org/10.3390/cancers13164224: 79 patients
Markowiak T, Ansari MKA, Neu R, Schalke B, Marx A, Hofmann H-S, Ried M. Evaluation of Surgical Therapy in Advanced Thymic Tumors. Cancers. 2021; 13(18):4516. https://doi.org/10.3390/cancers13184516: 73 patients
Riva G, Cavallo I, Gandini S, Ingargiola R, Pecorilla M, Imparato S, Rossi E, Mirandola A, Ciocca M, Orlandi E, Iannalfi A. Particle Radiotherapy for Skull Base Chondrosarcoma: A Clinical Series from Italian National Center for Oncological Hadrontherapy. Cancers. 2021; 13(17):4423. https://doi.org/10.3390/cancers13174423: 48 patients
Fouquet G, Wartski M, Dechmi A, Willems L, Deau-Fischer B, Franchi P, Descroocq J, Deschamps P, Blanc-Autran E, Clerc J, Bouscary D, Barreau S, Chapuis N, Vignon M, Cottereau A-S. Prognostic Value of FDG-PET/CT Parameters in Patients with Relapse/Refractory Multiple Myeloma before Anti-CD38 Based Therapy. Cancers. 2021; 13(17):4323. https://doi.org/10.3390/cancers13174323: 38 patients
Billingy NE, Tromp VNMF, van den Hurk CJG, Becker-Commissaris A, Walraven I. Health-Related Quality of Life and Survival in Metastasized Non-Small Cell Lung Cancer Patients with and without a Targetable Driver Mutation. Cancers. 2021; 13(17):4282. https://doi.org/10.3390/cancers13174282: 81 patients
Hartrampf PE, Lapa C, Serfling SE, Buck AK, Seitz AK, Meyer PT, Ruf J, Michalski K. Development of Discordant Hypermetabolic Prostate Cancer Lesions in the Course of [177Lu]PSMA Radioligand Therapy and Their Possible Influence on Patient Outcome. Cancers. 2021; 13(17):4270. https://doi.org/10.3390/cancers13174270: 32 patients
Kim HJ, Chang H-S, Ryu YH. Prognostic Role of Pre-Treatment [18F]FDG PET/CT in Patients with Anaplastic Thyroid Cancer. Cancers. 2021; 13(16):4228. https://doi.org/10.3390/cancers13164228: 40 patients
Kobashi Y, Uchiyama M, Matsui J. The “K-Sign”—A Novel CT Finding Suggestive before the Appearance of Pancreatic Cancer. Cancers. 2021; 13(16):4222. https://doi.org/10.3390/cancers13164222: 41 patients
Schroyen G, Blommaert J, van Weehaeghe D, Sleurs C, Vandenbulcke M, Dedoncker N, Hatse S, Goris A, Koole M, Smeets A, van Laere K, Sunaert S, Deprez S. Neuroinflammation and Its Association with Cognition, Neuronal Markers and Peripheral Inflammation after Chemotherapy for Breast Cancer. Cancers. 2021; 13(16):4198. https://doi.org/10.3390/cancers13164198: 74 patients
Katagiri K, Shiga K, Ikeda A, Saito D, Oikawa S-i, Tsuchida K, Miyaguchi J, Kusaka T, Tamura A, Nakayama M, Izumisawa M, Yoshida K, Ogasawara K, Takahashi F. The Influence of Young Age on Difficulties in the Surgical Resection of Carotid Body Tumors. Cancers. 2021; 13(18):4565. https://doi.org/10.3390/cancers13184565: 20 patients
Numoto I, Tsurusaki M, Oda T, Yagyu Y, Ishii K, Murakami T. Transcatheter Arterial Embolization Treatment for Bleeding Visceral Artery Pseudoaneurysms in Patients with Pancreatitis or following Pancreatic Surgery. Cancers. 2020; 12(10):2733. https://doi.org/10.3390/cancers12102733: 42 patients
According to your comment, the relatively small number of patients has been considered in the paragraph 4.11. as follows:
4.11. Strengths and Limitations
In this study, cross-sectional associations between nutritional, clinical, and inflammatory parameters were widely analyzed. There are not a lot of similar studies in the world literature. Our analysis is comprehensive and involves numerous parameters. To our knowledge, this is the first study regarding the association between NS and tumor location in PC patients. This study has got some limitations. This is a retrospective, single center study with a relatively small number of patients. Further prospective, multicenter, large studies are needed to validate our observations.
Despite of publications related to similar subject, there are not numerous publications including such comprehensive analysis of various nutritional and immune parameters and clinicopathological factors in PC patients. In our study, we performed a comprehensive analysis including combined nutritional and immune parameters. We analyzed not only prognostic values of nutritional parameters but also complex associations between them and clinicopathological patient- and tumor related factors. In addition, we compared these parameters between two location of pancreatic cancer (proximal and distal). We have not found such precise comparison regarding tumor location in previous publications. To our knowledge, this is the first such study on this subject.
Moreover, our study is a Polish / Central European voice in the worldwide discussion regarding this subject. It may be used in a further meta-analysis. So far, there are only singular meta-analyses regarding the prognostic role of the hematological and immune parameters in PC patients. So far, complex associations between nutritional parameters and clinicopathological factors have been not clearly determined.
In conclusion, our study included a detailed and comprehensive comparative analysis of numerous nutritional and immune parameters. Taking into account all the above-mentioned arguments, the novelty of our study is considerable.
According to your suggestions, conclusions of our study have been modified as follows:
Our study confirmed that nutritional impairment correlates with systemic inflammatory response in PC patients. Obesity (BMI≥30 kg/m2) and malnutrition (NRS 2002≥3) predict postoperative complications. On the other hand, higher postoperative morbidity rate is associated with longer duration of hospitalization. Therefore, assessment of nutritional and immune status using basic diagnostic tools and PNI and immune ratio (NLR, MLR, PLR) calculation should be the standard management of PC patients before surgery to improve the postoperative outcome.
In our opinion, patients with deteriorated nutritional and immune parameters need particular attention during perioperative management. Generally, a longer preoperative nutritional support may be required for PC patients at nutritional risk. Moreover, because of higher WL in patients with the tumor located within the pancreatic head, these patients may require more careful nutritional support in the perioperative period. In addition, due to lower levels of laboratory nutritional parameters, older patients (> 65 years) also need more careful nutritional monitoring and intervention.

Reviewer 3 Report
This manuscript describes about the association between nutritional or immune status and postoperative complications in patients with pancreatic cancer. This manuscript states some important issues regarding nutritional and immune status in patients with pancreatic cancer. However further discussion would be required to confirm this conclusion. Therefore, authors would clarify several matters that mentioned below.
- Authors reported that laboratory blood tests were performed at hospital admission. However, some patients had neoadjuvant therapy. Neoadjuvant therapy may cause hematologic toxicity because of its myelosuppressive effect. Systemic immune inflammatory markers before neoadjuvant therapy might not precisely reflect risk of postoperative complications. How did neoadjuvant therapy affect systemic immune inflammatory markers after neoadjuvant therapy? Authors could evaluate the association between postoperative complications and systemic immune inflammatory markers after neoadjuvant therapy.
- Authors reported that body mass index (BMI) ≥30 and NRS 2002 ≤3 predicted postoperative complications. In contrast, some studies have reported that malnutrition is a risk factor of postoperative complications. Here is a discrepancy. Authors could evaluate whether obese or malnutrition adversely affects postoperative complications in this study.
- In this study, higher NRS 2002 score was linked with the longer duration of hospitalization and NRS 2002 ≤3 predicted postoperative complications. Postoperative complications seem to prolong duration of hospitalization. What is the definition of discharge in this study? Was there the correlation between duration of hospitalization and the incidence of postoperative complications?
Author Response
Dear Editors,
Dear Reviewers,
Thank you for peer reviewing of our manuscript cancers-1367776, entitled " Associations between Nutritional and Immune Status and Clinicopathologic Factors in Patients with Pancreatic Cancer: a Comprehensive Analysis".
Thank you for your questions and comments. We have fully addressed all the comments and my responses appear below. Our revised work includes corrections according to reviewers’ comments in the text. Our revisions, made according to reviewers’ comments, are highlighted in the green print.
We take this opportunity to express my gratitude to the reviewers for their constructive and useful remarks. Their comments allowed us to identify areas in my manuscript that needed modification.
We also thank you for allowing me to resubmit a revised copy of the manuscript.
We hope that the revised manuscript is now acceptable for publication in Cancers.
Yours sincerely,
Beata Jabłońska.
Responses to Reviewers’ comments
Reviewer 3
This manuscript describes about the association between nutritional or immune status and postoperative complications in patients with pancreatic cancer. This manuscript states some important issues regarding nutritional and immune status in patients with pancreatic cancer. However further discussion would be required to confirm this conclusion. Therefore, authors would clarify several matters that mentioned below.
Comment:
- Authors reported that laboratory blood tests were performed at hospital admission. However, some patients had neoadjuvant therapy. Neoadjuvant therapy may cause hematologic toxicity because of its myelosuppressive effect. Systemic immune inflammatory markers before neoadjuvant therapy might not precisely reflect risk of postoperative complications. How did neoadjuvant therapy affect systemic immune inflammatory markers after neoadjuvant therapy? Authors could evaluate the association between postoperative complications and systemic immune inflammatory markers after neoadjuvant therapy.
Answer:
Thank you for your comment. Blood tests were performed at hospital admission in the Department of Digestive Tract Surgery, one day before surgery. Therefore, patients have already received and finished neoadjuvant therapy and presented laboratory results have been taken already after neoadjuvant chemotherapy. So, these parameters reflect myelosuppressive effect of neoadjuvant chemotherapy and the evaluated association between postoperative complications and systemic immune inflammatory markers regards parameters after neoadjuvant therapy in patients who had received it. The blood tests before neoadjuvant chemotherapy have been taken in oncological units before oncological treatment. The aim of our study was to assess preoperative nutritional and immune parameters taken directly surgery. We have compared nutritional and immune parameters depending on presence of postoperative complications in Table S9 Table S9 Comparison of selected clinicopathological and nutritional parameters depending on presence of postoperative complications.
According to your suggestion, we have added comparison of rate of neoadjuvant chemotherapy in Table S9 (Supplementary materials) as follows:
|
Table S9 Comparison of selected clinicopathological and nutritional parameters depending on the presence of postoperative complications |
|||
|
Feature |
No complications |
Complications |
P value |
|
Neoadjuvant chemotherapy No Yes |
54 (87.10 %) 8 (12.90%) |
14 (77.78 %) 4 (22.22 %) |
0.2645 |
Although the rate of the neoadjuvant chemotherapy in the complication group was almost two folds higher compared to the patients without complications (22.22% vs. 12.90%), the difference was not statistically significant (p=0.3297).
The mentioned above comment has been added in the manuscript (paragraph 3.5.).
According to your suggestion, we have added a new table Table S11 Comparison of selected clinicopathological, immune and nutritional parameters depending on the use of neaoadjuvant chemotherapy. In this table, clinicopathological as well nutritional and immune parameters have been compared between patients with and without neoadjuvant chemotherapy. This table reflects the impact of neoadjuvant chemotherapy on compared parameters.
|
Table S11 Comparison of selected clinicopathological, immune and nutritional parameters depending on the use of neoadjuvant chemotherapy |
|||
|
Feature |
No neoadjuvant chemotherapy (n=68) |
Neoadjuvant chemotherapy (n=12) |
P value |
|
Age |
65.28±8.60 |
66.33±7.25 |
0.6904 |
|
Gender |
34 (50.00 %) Male 34 (50.00 %) Female |
6 (50.00 %) Male 6 (50.00 %) Female |
1.0000 |
|
Weight |
71.02±12.88 |
70.33±13.69 |
0.8660 |
|
Weight loss |
7.03±8.22 |
11.57±8.44 |
0.1901 |
|
BMI |
25.16±3.53 |
24.37±3.44 |
0.4762 |
|
BMI groups <18.5 ≥18.5 |
3 (4.41 %) 65 (95.59 %) |
12 (100.00 %) 0 (0.00 %) |
0.0000001 |
|
BMI groups <30 ≥30 |
63 (92.65 %) 5 (7.35 %) |
12 (100.00 %) 0 (0.00 %) |
0.3320 |
|
NRS 2002 |
2.03±1.10 (1-5) |
2.75±0.96 (1-4) |
0.0375 |
|
NRS 2002 groups 1. < 3 2. ≥ 3 |
54 (79.41 %) 14 (20.59 %) |
5 (41.67 %) 7 (58.33 %) |
0.0061 |
|
Tumor location
|
49 (72.06 %) Proximal 19 (27.94 %) Distal |
9 (75.00%) Proximal 3 (25.00 %) Distal |
0.8334 |
|
Tumor depth (T) |
9 (13.24 %) T1 51 (75.00 %) T2 8 (11.76 %) T3
28 (41.18 %) T1-2 40 (58.82 %) T3 |
4 (33.33 %) T1 7 (58.33 %) T2 1 (8.33 %) T3
5 (41.67 %) T1-2 7 (58.33 %) T3 |
0.2195
0.9746 |
|
Lymph node invasion (N) |
9 (13.24 %) N0 18 (26.47 %) N1 41 (60.29 %) N2
9 (13.24 %) N0 59 (86.76 %) N+ |
0 (0.00 %) N0 4 (33.33 %) N1 8 (66.67 %) N2
0 (0.00 %) N0 12 (100.00 %) N+ |
0.4004
0.1810 |
|
Distal metastasis (M) |
62 (91.18 %) 6 (8.82 %) |
10 (83.33 %) 2 (16.67 %) |
0.4037 |
|
Histological type Adenocarcinoma Adenosquamous carcinoma |
63 (92.65 %) 5 (7.35 %) |
12 (100.00 %) 0 (0.00 %) |
0.3320 |
|
Histological grading
|
7 (10.29 %) G1 42 (61.76 %) G2 19 (27.94 %) G3
|
0 (8.33 %) G1 15 (66.67 %) G2 6 (25.00 %) G3 |
0.9462 |
|
Duration of postoperative hospitalization [days] < 10 > 10 |
34 (50.00 %) 34 (50.00 %) |
8 (66.67 %) 4 (33.33 %) |
0.2865 |
|
Duration of hospitalization in ICU [days] |
4±6.23 (0-22) |
2±0.00 (2-2) |
0.7624 |
|
Postoperative parenteral nutrition |
12 (17.65 %) |
2 (16.67 %) |
0.9343 |
|
Duration of parenteral nutrition [days] |
6.45±5.59 (0-15) |
11±15.56 (0-22) |
0.3537 |
|
Postoperative complication rate |
14 (20.59 %) |
4 (33.33 %) |
0.3297 |
|
Postoperative complications according to Clavien Dindo classification 0 1 2 3 4 5 |
53 (77.94 %) 5 (7.35 %) 1 (1.47 %) 8 (11.76 %) 0 (0.00 %) 1 (1.47 %) |
9 (75.00 %) 2 (16.67 %) 1 (8.33 %) 0 (0.00 %) 0 (0.00 %) 0 (0.00 %) |
0.3387 |
|
Reoperation rate |
8 (11.76 %) |
1 (8.33 %) |
0.7287 |
|
Mortality rate |
1 (1.47 %) |
0 (0.00 %) |
0.6725 |
|
Total protein [g/dl] |
6.04±0.97 |
6.44±0.89 |
0.1933 |
|
Albumin [g/dl] |
3.48±0.74 |
3.64±0.71 |
0.4875 |
|
Hemoglobin [g/dl] |
13.15±1.54 |
12.71±1.46 |
0.3565 |
|
CRP [mg/l] |
7.71±11.37 |
4.58±3.48 |
0.4195 |
|
White blood cell count [/mm3] |
7.25±1.99 |
8.23±2.95 |
0.1490 |
|
Total lymphocyte count [/mm3] |
1.91±0.77 |
2.15±0.81 |
0.4440 |
|
Neutrophil count [/mm3] |
4.42±1.60 |
4.70±1.77 |
0.6668 |
|
Monocyte count [/mm3] |
0.60±0.22 |
0.76±0.38 |
0.1128 |
|
Platelet count [/mm3] |
262.71±107.39 |
213.00±59.15 |
0.1238 |
|
PNI |
44.78±8.43 |
46.87±8.76 |
0.5426 |
|
NLR |
2.56±1.36 |
2.89±2.30 |
0.5970 |
|
MLR |
0.36±0.19 |
0.38±0.16 |
0.7976 |
|
PLR |
160.28±96.13 |
131.54±96.94 |
0.4619 |
|
NRS 2002, Nutritional Risk Score; BMI, body mass index; ASA, American Society of Anesthesiologists; CRP, C-reactive protein; PNI, prognostic nutritional index; NLR, Neutrophil/lymphocyte ratio; MLR, Monocyte/lymphocyte ratio; PLR, Platelet/lymphocyte ratio. Significant results (p<0.05) are highlighted in red print. |
|||
Also a new paragraph presenting this subject has been added as follows:
- Comparison of selected clinicopathological factors as well as immune and nutritional parameters depending on the use of neoadjuvant chemotherapy
In the divison of our cohort depending on the use of neoadjuvant chemotherapy, we did not find statistical differences in immune parameters and postoperative complications between patients without and with neoadjuvant chemotherapy. It may partly be associated with the fact that only 12/80 patients had received neoadjuvant chemotherapy in our study. There was statistical difference in NRS 2002 score between two subgroups (2.0294 vs. 2.75 points; p=0.037461), in patients without and with neoadjuvant chemotherapy, respectively), because neoadjuvant chemotherapy is one of NRS 2002 determinants. The interesting observed phenomenon in our cohort was that all patients, who had received neoadjuvant chemotherapy, and only 4.41% of patients without neoadjuvant chemotherapy, had BMI<18.5 kg/m2 (p=0.0000001). The postoperative morbidity, reoperation, and mortality rates were similar in both groups (p>0.05). Although the rate of neoadjuvant chemotherapy in the complication group was almost two folds higher compared to the patients without complications (22.22% vs. 12.90%), the difference was not statistically significant (p=0.3297) (Table 9). In addition, we compared postoperative complications according to Clavien Dindo classification between both groups. There was no statistical difference between two groups (p>0.05). All comparisons between the two group are presented in Table S11.
In this table, we have also compared the rate of postoperative complications, reoperations, and mortality between patients without and with neoadjuvant chemotherapy. The morbidity, reoperation, and mortality rates were similar in both groups (p>0.05). In addition, we have compared postoperative complications according to Clavien Dindo classification between both groups. There was no statistical difference between two groups (p>0.05).
Also, according to your suggestion, we have added neoadjuvant chemotherapy to the logistic regression analysis concerning prediction of postoperative complications (Table 5) as follows:
|
Table 5 Significant predictors of postoperative complications in multiple binomial logistic regression analysis |
|||||||||
|
|
b |
SE b |
-95% CI |
+95% CI |
Wald stat. |
p value |
OR |
-95% CI |
+95% CI |
|
Intercept |
-3.903865 |
1.191759 |
-6.23967 |
-1.56806 |
10.730317 |
0.001054 |
0.020164 |
0.00195 |
0.208449 |
|
BMI groups <30 vs. ≥30 |
2.154665 |
0.989986 |
0.214328 |
4.095002 |
4.736977 |
0.029521 |
8.625 |
1.239029 |
60.03947 |
|
NRS 2002 groups <3 vs. ≥3 |
1.056053 |
0.600853 |
-0.121598 |
2.233704 |
3.089115 |
0.048818 |
2.875 |
0.885504 |
9.334375 |
|
Neoadjuvant chemotherapy 0 vs. 1 |
-0.60343 |
1.214431 |
-2.98367 |
1.776811 |
0.246892 |
0.619272 |
0.546933 |
0.050607 |
5.910974 |
Also in this analysis, Neoadjuvant chemotherapy has not predicted postoperative complications (p>0.05).
In conclusion, we have not found statistical differences in immune parameters and postoperative complications between patients without and with neoadjuvant chemotherapy. It may partly be associated with the fact that only 12/80 patients had received neoadjuvant chemotherapy in our study. There was only statistical difference in NRS 2002 score between two subgroups (2.0294 vs. 2.75 points; p=0.037461, in patients without and with neoadjuvant chemotherapy, respectively), because neoadjuvant chemotherapy is one of NRS 2002 determinants. This observation has been discussed in the paragraph 4.10. as follows:
As we mentioned above, 33% of patients at nutritional risk vs. 8% in patients without nutritional risk received neoadjuvant chemotherapy which confirms the fact that preoperative chemotherapy aggravates NS. In our study, we have not found statistical differences in immune parameters and postoperative complications between patients without and with neoadjuvant chemotherapy. It may partly be associated with the fact that only 12/80 patients had received neoadjuvant chemotherapy in our study. There was statistical difference in NRS 2002 score between two subgroups in patients without and with neoadjuvant chemotherapy, because neoadjuvant chemotherapy is one of NRS 2002 determinants. The interesting phenomenon in our cohort was that all patients who had received neoadjuvant chemotherapy and only 4.41% of patients without neoadjuvant chemotherapy had BMI<18.5 kg/m2. This difference was strongly significant. The problem of neoadjuvant therapy in PC patients undergoing surgery was discussed by Kimura et al [48]. In this retrospective study including 199 patients with borderline resectable PC, the authors showed no association between the high / low preoperative PNI and postoperative complications and significantly better prognosis in patients with high (> 42.50) preoperative PNI [44]. So far, the impact of neoadjuvant chemotherapy on the NS and postoperative complications in PC patients has been not clearly recognized and there is the limited number of relevant studies. Further studies are needed to answer this problem. Despite this fact, in our opinion, it is important to identify patients at nutritional risk following chemotherapy and to introduce nutritional intervention to improve their prognosis.
Comment:
- Authors reported that body mass index (BMI) ≥30 and NRS 2002 ≤3 predicted postoperative complications. In contrast, some studies have reported that malnutrition is a risk factor of postoperative complications. Here is a discrepancy. Authors could evaluate whether obese or malnutrition adversely affects postoperative complications in this study.
Answer:
Thank you for your comment. In our study, body mass index (BMI) ≥30 and NRS 2002 ≥3 predicted postoperative complications. We apologise, there was a typo in the phrase „NRS 2002 ≥3”, and it should be „NRS 2002 ≥3”. Thank to you very much for taking attention for this paragraph. It has been changed in the current manuscript as follows:
In multiple logistic regression analysis, BMI ≥30 kg/m2 (OR: 8.62; 95%CI: 1.24-60.04; p=0.029521) and NRS 2002 ≥3 (OR: 2.87; 95%CI: 0.88-9.33; p=0.048818) predicted postoperative complications (Table 5).
It is in accordance with the Table 5 (It was correct and it has been not changed). In this table we evaluated that both obese and malnutrition adversely affect postoperative complications in our study. So, our results are in accordance with the literature data.
|
Table 5 Significant predictors of postoperative complications in multiple binomial logistic regression analysis |
|||||||||
|
|
b |
SE b |
-95% CI |
+95% CI |
Wald stat. |
p value |
OR |
-95% CI |
+95% CI |
|
Intercept |
-3.903865 |
1.191759 |
-6.23967 |
-1.56806 |
10.730317 |
0.001054 |
0.020164 |
0.00195 |
0.208449 |
|
BMI groups <30 vs. ≥30 |
2.154665 |
0.989986 |
0.214328 |
4.095002 |
4.736977 |
0.029521 |
8.625 |
1.239029 |
60.03947 |
|
NRS 2002 groups <3 vs. ≥3 |
1.056053 |
0.600853 |
-0.121598 |
2.233704 |
3.089115 |
0.048818 |
2.875 |
0.885504 |
9.334375 |
|
Neoadjuvant chemotherapy 0 vs. 1 |
-0.60343 |
1.214431 |
-2.98367 |
1.776811 |
0.246892 |
0.619272 |
0.546933 |
0.050607 |
5.910974 |
Our results have been discussed and compared with the worldwide literature in the discussion (paragrap 4.1., and paragraph 4.2.) as follows:
Moreover, obesity is associated with higher postoperative morbidity and complication rate that was confirmed in our study, in which the percentage of obese patients was higher in the complication group as well as BMI ≥30 was a risk factor for postoperative morbidity.
In our cohort, the higher NRS 2002 classification, as a marker of malnutrition, and mentioned above obesity were associated with a higher risk of postoperative complications. The higher NRS 2002 score was also associated with significantly longer duration of hospitalization in our cohort. Moreover, the incidence of postoperative complications correlated with longer hospital stay in our patients.
Comment:
- In this study, higher NRS 2002 score was linked with the longer duration of hospitalization and NRS 2002 ≤3 predicted postoperative complications. Postoperative complications seem to prolong duration of hospitalization. What is the definition of discharge in this study? Was there the correlation between duration of hospitalization and the incidence of postoperative complications?
Answer:
Thank you for your comment. We have evaluated the significant association between duration of hospitalization and the incidence of postoperative complications in the Table S9 presented in the Supplementary materials as follows:
|
Table S9 Comparison of selected clinicopathological and nutritional parameters depending on the presence of postoperative complications |
|||
|
Feature |
No complications |
Complications |
P value |
|
Duration of hospitalization [days] |
12.17±3.56 |
22.94±15.05 |
0.0001 |
|
Duration of hospitalization in ICU [days] |
1.33±1.00 |
10.25±8.09 |
0.0087 |
There was significantly longer duration of hospitalization in the complication group (p=0.0001).
According to your suggestion, we have added comparison of duration of postoperative hospitalization according to cuttof 10 days between patients with and without complications (Table S9). The longer duration of hospitalization (> 10 days after surgery) was significantly more frequently reported in the complication group compared to non-complication group (77.78 % vs. 38.71 %; p=0.0035, respectively).
|
Table S9 Comparison of selected clinicopathological and nutritional parameters depending on the presence of postoperative complications |
|||
|
Feature |
No complications |
Complications |
P value |
|
Duration of postoperative hospitalization (patients number) < 10 days > 10 days |
38 (61.29 %) 24 (38.71 %) |
4 (22.22 %) 14 (77.78 %) |
0.0035 |
This report has been added in results section (paragraph 3.5.) as follows:
In addition, we compared duration of postoperative hospitalization according to cuttof 10 days (median value) between patients with and without complications (Table S9). The longer duration of hospitalization (> 10 days after surgery) was significantly more frequently reported in the complication group compared to non-complication group (77.78 % vs. 38.71 %; p=0.0035, respectively). On the other side, the significantly higher complication rate was noted in patients hospitalized more than 10 days compared to the shorter hospital stay (36.84% vs. 9.52 %; p=0.0035, respectively). Also, in Spearman correlation, duration of postoperative hospitalization was positively correlated with postoperative complication rate (r = 0.33, p = 0.0031). All comparisons between the two group are presented in Table S9 and S10.
According to your suggestion, we have analyzed association between postoperative complication rate and duration of postoperative hospitalization using multiple linear regression analysis in Table 7 as follows:
|
Table 7 Association between complication rate and duration of postoperative hospitalization in multiple regression linear analysis |
|||||
|
Model 1 |
Complications R= 0.32669159 R^2= 0.10672739 Adjusted R2= 0.09527518 F(1.78)=9.3194 p<0.00310 Standard Error of estimate: 0.39970 |
||||
|
|
b |
Standard Error of b |
p value |
r |
p value |
|
Intercept |
0.095238 |
0.107015 |
0.126588 |
|
|
|
Duration of postoperative hospitalization ≤10 vs. >10 days |
0.273183 |
0.089487 |
0.003100 |
0.326691 |
0.003100 |
It has been described in results section (paragraph 3.7.) as follows:
In the multiple linear regression analysis, higher NRS 2002 score was linked with the longer duration of hospitalization (b = 7.67948; p=0.043816) (Table 6), and longer duration of postoperative hospitalization was associated with higher complication rate (b = 0.273183; p=0.003100) (Table 7).
This observation has been discussed in discussion section (paragraph 4.2.) as follows:
The higher NRS 2002 score was also associated with significantly longer duration of hospitalization in our cohort. Moreover, the incidence of postoperative complications correlated with longer hospital stay in our patients.
In our study, the patient was discharged when he no longer needed to receive inpatient care and could go home, precisely when the oral diet was introduced, with proper wound healing, and without presence of postoperative complications or if postoperative complications have been already successfully treated.
According to your suggestion, we have added comparison of complication rate in two subgroups divided according to median duration of hospitalization (10 days) in Table S10 as follows:
|
Table S10 Comparison of the incidence of postoperative complications according to duration of postoperative hospitalization |
|||
|
Duration of postoperative hospitalization |
< 10 days (n=42) |
> 10 days (n=38) |
P value |
|
Complications No Yes |
38 (90.48 %) 4 (9.52 %) |
24 (63.16 %) 14 (36.84 %) |
0.0035 |
|
Significant results (p<0.05) are highlighted in red print. |
|||
The significantly higher complication rate has been noted in patients hospitalized more than 10 days compared to the shorter hospital stay (36.84% vs. 9.52 %; p=0.0035, respectively) (Table S10).

Round 2
Reviewer 3 Report
This manuscript is revised well according to reviewer's comments . Authors seems to clarify the association between nutritional and immune status and pancreatic cancer in this study. I have no additional comments.
This manuscript is a resubmission of an earlier submission. The following is a list of the peer review reports and author responses from that submission.